# Going beyond Integration: The Emerging Role of HIV-1 Integrase in Virion Morphogenesis

**DOI:** 10.3390/v12091005

**Published:** 2020-09-09

**Authors:** Jennifer L. Elliott, Sebla B. Kutluay

**Affiliations:** Department of Molecular Microbiology, Washington University School of Medicine, Saint Louis, MO 63110, USA; elliott.j@wustl.edu

**Keywords:** HIV-1, integrase, maturation, integrase–RNA interactions, protein–RNA interactions

## Abstract

The HIV-1 integrase enzyme (IN) plays a critical role in the viral life cycle by integrating the reverse-transcribed viral DNA into the host chromosome. This function of IN has been well studied, and the knowledge gained has informed the design of small molecule inhibitors that now form key components of antiretroviral therapy regimens. Recent discoveries unveiled that IN has an under-studied yet equally vital second function in human immunodeficiency virus type 1 (HIV-1) replication. This involves IN binding to the viral RNA genome in virions, which is necessary for proper virion maturation and morphogenesis. Inhibition of IN binding to the viral RNA genome results in mislocalization of the viral genome inside the virus particle, and its premature exposure and degradation in target cells. The roles of IN in integration and virion morphogenesis share a number of common elements, including interaction with viral nucleic acids and assembly of higher-order IN multimers. Herein we describe these two functions of IN within the context of the HIV-1 life cycle, how IN binding to the viral genome is coordinated by the major structural protein, Gag, and discuss the value of targeting the second role of IN in virion morphogenesis.

## 1. Introduction

Human immunodeficiency virus type 1 (HIV-1) is the causative agent of AIDS, and since its discovery in 1983 [1,2] has become one of the leading causes of the death worldwide due to infectious disease. Intensive study of the HIV-1 life cycle has led to the identification of viral enzymes essential for virus replication, and antiretroviral compounds that specifically inhibit the functions of these enzymes have transformed HIV-1 infection from a death sentence into a manageable disease. The HIV-1 integrase enzyme (IN) plays a vital role in the viral life cycle by catalyzing the integration of viral DNA into the host chromosome. This function has been successfully targeted by a class of antiretrovirals known as integrase strand-transfer inhibitors (INSTIs) [3]. Four FDA-approved INSTIs, raltegravir [4], elvitegravir [5], dolutegravir [6], and bictegravir [7], have become key components of anti-retroviral therapy regimens and are both highly effective and well tolerated ([8,9,10], reviewed in [11]). A fifth, cabotegravir [12], is currently in late stage clinical trials. However, despite high barriers with the second-generation INSTIs, treatment does select for drug resistance [13,14,15], and mutations conferring resistance to multiple INSTIs have been reported in clinical settings [16,17], highlighting the need for continued research and development of both improved and novel antiretroviral compounds.

It was recently discovered that IN has a second essential role in the HIV-1 life cycle. IN binds the viral RNA (vRNA) genome in virions and is necessary for the proper placement of vRNA within the viral capsid lattice during virion maturation [18]. Loss of IN–RNA binding leads to mislocalization of the viral genome in virions and prevents viral replication in target cells [18]. This discovery opens up new avenues for therapeutic targeting of the second function of IN that is independent of its already targeted catalytic function. The purpose of this review is to provide an overview of the multiple roles of IN in the HIV-1 life cycle, with a focus on its function during virion maturation and morphogenesis. In addition, the value of targeting virion morphogenesis as a therapeutic strategy will be discussed.

## 2. Overview of the HIV-1 Life Cycle

The HIV-1 life cycle can be broadly divided into an early stage (up to integration) and a late stage (after integration). Mature HIV-1 virions consist of two copies of a single-stranded RNA genome and replicative enzymes (i.e., reverse transcriptase (RT) and IN) encased in a conical protein lattice made up by the viral capsid (CA) protein, together forming the viral “core” (Figure 1). The viral genome inside the core exists in the form of a viral ribonucleoprotein complex (vRNP) bound and condensed by the viral nucleocapsid (NC) protein, and associated with RT and IN enzymes [19]. The viral core itself is enclosed within the viral lipid envelope derived from the host cell plasma membrane. The surface of the virion is studded with the viral envelope (Env) glycoprotein trimers (reviewed in [20]). During viral entry, the Env glycoproteins engage the CD4 receptor and CXCR4/CCR5 coreceptors on the target cell, which induces a series of conformational changes in Env, resulting in membrane fusion and release of the viral core into the cytoplasm (reviewed in [21,22]).

After entry, the viral core is transported towards the nucleus along microtubules [23,24] and reverse transcription ensues (Figure 1). During this stage, the core undergoes an uncoating process in which the capsid disassembles and CA monomers are shed from the lattice (reviewed in [25]). While uncoating and reverse transcription have long been thought to occur in the cytoplasm, recent studies have provided evidence that these processes are not completed until after nuclear entry [26,27]. Notwithstanding, the timing and degree of uncoating is critical for completion of reverse transcription. Several mutations in CA can destabilize the CA lattice, resulting in severe defects in reverse transcription [28,29,30,31], presumably due to premature degradation of the core components, including IN [32]. IN remains associated with the reverse transcription complex, and following completion of cDNA synthesis, a multimer of IN binds to both ends of the linear viral DNA to form the intasome, or the stable synaptic complex. The reverse transcription complex is actively transported into the nucleus through the nuclear pore complexes involving Nup 358, Nup 153, as well as CPSF6 (reviewed in [33,34]), and upon entering the nucleus IN catalyzes the integration of the viral DNA into the host cell chromosome (reviewed in [35]).

After the viral DNA is integrated into the host chromosome it serves as a template from which single full-length viral mRNA is transcribed by the host RNA polymerase II machinery (reviewed in [36], Figure 1). This viral transcript can remain unspliced or undergo a complex series of splicing events (reviewed in [36]). Fully spliced HIV-1 mRNAs code for the regulatory Tat, Rev, and Nef proteins and are exported from the nucleus via the NXF1/NXT1 pathway (reviewed in [37,38,39]). Partially spliced mRNAs code for the viral envelope Env and accessory proteins Vif, Vpr, and Vpu, while the unspliced full-length HIV-1 mRNAs can be packaged into virions as the genomic RNA or translated to generate the major structural protein, Gag, and the frameshifting variant Gag–Pol polyprotein [40,41], which additionally codes for the replicative enzymes protease (PR), RT, and IN (reviewed in [37,38,39]). Both partially spliced and unspliced HIV-1 RNAs are retained in the nucleus until they can be exported by the viral Rev protein [42,43] through a CRM1-dependent pathway [44,45,46].

Unspliced dimeric vRNA is trafficked to the plasma membrane by Gag, and this complex subsequently nucleates the assembly of nascent virions (reviewed in [47,48,49], Figure 1). During this process, the Gag and Gag–Pol polyproteins polymerize around the vRNA, acquire Env glycoproteins recruited to the budding site, and virions bud off from the infected cell in an immature state. During or shortly after budding, the virion undergoes a maturation process in which the Gag and Gag–Pol polyproteins are cleaved into separate mature proteins by the virally encoded PR enzyme. This triggers a structural rearrangement within the virion, whereby the cleaved NC proteins condense the vRNA together with RT and IN to form the vRNP, the viral CA lattice assembles around the vRNP, and the now mature virion is ready to infect a new target cell and reinitiate the viral life cycle (reviewed in [47,48]).

## 3. The Essential Catalytic Function of Integrase

A defining feature of the retroviral life cycle is integration of the reverse-transcribed viral DNA into the host chromosome. During integration, a multimer of IN binds either end of the linear viral DNA to form the intasome complex, which inserts the reverse transcribed viral DNA into the host DNA (reviewed in [35,50]). This function of IN was identified shortly after the discovery of HIV-1 in the early 1980s [51,52,53,54], and has been extensively studied. Each IN molecule is composed of three functionally distinct domains: an N-terminal domain (NTD), a catalytic core domain (CCD), and a C-terminal domain (CTD). The NTD and CTD domains mediate DNA binding and play important structural roles in the intasome complex, while the CCD contains a highly conserved D,D,-35-E motif in the enzyme active site necessary for catalytic activity (reviewed in [35,50]). Mutations at these conserved residues, collectively referred to as class I IN mutations [55,56,57], predictably abolish the catalytic activity of IN in vitro [55] and block the viral life cycle at an integration stage in infected cells [57]. 

Shortly after nuclear entry and prior to integration, viral DNAs are rapidly loaded with core as well as linker histones [58,59,60]. Recent studies revealed that in contrast to integrated proviral DNAs, unintegrated retroviral DNAs are transcriptionally silenced through deposition of histones bearing repressive transcription marks [58,59,60]. In line with this, treatment of infected cells by histone deacetylase inhibitors can allow viral gene expression from unintegrated retroviral DNAs [58,59,61,62] by preventing the removal of histone H3 acetylation [58,59], an epigenetic modification associated with active transcription. Silencing of the unintegrated retroviral DNAs is mediated by the HUSH complex [59], which epigenetically represses genes by recruiting the H3K9me3 methyltransferase SET domain bifurcated 1 (SETDB1) [63]. Depletion of a component of the HUSH complex, NP220, allows viral gene expression from the unintegrated DNAs of some retroviruses [59]. Retroviruses appear to have evolved strategies to prevent silencing by the HUSH complex. For example, the HIV-2 and SIV Vpr/Vpx proteins induce the proteasomal degradation of HUSH, allowing for enhanced viral gene expression [64,65]. Finally, expressing the Tax transcription factor encoded by human T-cell leukemia virus 1 (HTLV-1) or the ICP0 immediate early protein of herpes simplex virus type-1 (HSV-1) in cells prevents or reverses the epigenetic silencing of integrated HIV-1 DNA, and can rescue the infectivity of an integration-defective HIV-1 virus [66], demonstrating that the usual requirement for integration can be bypassed if epigenetic silencing of unintegrated viral DNA is prevented.

IN is a dynamic protein and can form a population of monomers, dimers, and tetramers in vitro, and as noted above forms multimers during integration (reviewed in [67]). Early studies indicated that IN may function as a multimer by demonstrating that catalytically inactive mutant IN proteins bearing substitutions at different sites could trans-complement each other and regain catalytic activity in vitro [68,69,70]. More recently, a mechanistic study using a small molecule inhibitor found that the compound binds and selectively acetylates Lys173 at the interface between two dimers within an IN tetramer [71]. Importantly, the compound interfered with the interplay between IN subunits in a manner that correlated with its ability to inhibit IN catalytic activity, providing further evidence that proper IN multimerization is critical for its function [71].

The first retroviral intasome to be structurally characterized was the prototype foamy virus (PFV) from the spumavirus genus, which consists of a tetramer of IN made up of a dimer of dimers with viral DNA between the two subunits ([72,73] and reviewed in [74]). Each dimer includes an inner and outer IN molecule, with the inner subunits interacting with the viral and host DNA. The catalytic site in the inner IN CCD cooperatively coordinates the integration reaction with the NTD of the opposing inner IN, while the inner CTDs bind the host DNA and help hold the two dimers together. Meanwhile, the outer IN subunits further stabilize the complex by contacting the inner IN molecules at the CCD–CCD interface.

It was generally assumed that the HIV-1 intasome complex shared a similar structure. Tetrameric IN binds to viral DNA in cross-linking experiments [75], is catalytically active in vitro [76], and has been observed to interact with viral DNA by atomic force microscopy [77,78]. However, recent cryogenic electron microscopy (cryo-EM) studies have suggested that HIV-1 IN forms even higher-order multimers within the intasome complex [79]. In 2016 and 2017, several new retroviral intasomes were characterized in addition to HIV-1, including that of the β-retrovirus mouse mammary tumor virus (MMTV) [80], the α-retrovirus Rous sarcoma virus (RSV) [81], and the lentiviruses maedi-visna virus (MVV) [82]. Interestingly, there was a surprising diversity in the structure of the different retroviral intasomes (reviewed in [74]). While detailed structural analysis of the HIV-1 intasome has long been hindered by the propensity of HIV-1 IN to aggregate in solution, the aforementioned study overcame this issue by generating a hyper-active HIV-1 IN mutant protein with improved solubility. Single particle cryo-EM structures of the HIV-1 IN construct in complex with DNA indicate that a higher-order multimer of several tetramers may be needed to efficiently integrate viral DNA, although a lower-order intasome consisting of an IN tetramer was also observed [79].

During integration, IN multimers assemble at either end of the linear viral DNA to form the intasome complex and catalyze insertion of the viral DNA in two separate steps: 3’ processing and strand transfer. During 3’ processing, IN hydrolyzes a phosphodiester bond at either end of the viral DNA and removes two to three nucleotides in front of an invariant 5’-CA-3’ dinucleotide, creating free 3’ hydroxyl groups [35,83,84,85]. Then, during the strand transfer reaction, the intasome binds the target host DNA and uses the 3’ hydroxyls at either end of the viral DNA as nucleophiles to cut the host DNA in a staggered fashion, at the same time joining the viral DNA to the 5’ ends of the cut host DNA [86,87,88]. Finally, the intasome dissembles, leaving loose 5’ overhangs on the viral DNA and a pair of single-stranded gaps on either side of the integrated viral DNA, which are subsequently repaired by host cell machinery (reviewed in [89]). In an in vitro study using model DNA substrates that mimicked retroviral integration intermediates, the base-excision repair (BER) pathway enzymes DNA polymerase β, flap endonuclease 1 (FEN1), and ligase I were able to repair the gap and 5′ two-base overhangs [90]. SiRNA knockdown studies have similarly found that components of the BER pathway are important for retroviral integration [91,92]. As a result of integration and subsequent gap repair, a short segment of the target DNA sequence is duplicated and flanks the integrated provirus. The length of the duplicated sequence varies between retroviruses, with HIV-1 generating 5-bp duplications [93,94].

HIV-1 does not integrate randomly, but rather preferentially targets transcriptionally active genes in the nuclear periphery [95,96,97]. The pre-integration complex (PIC) is guided to its integration site by the chromatin-associated cellular protein lens epithelium-derived growth factor (LEDGF), also called transcriptional coactivator p75, which interacts with IN at its C-terminal integrase-binding domain [98,99,100,101]. The LEDGF/p75 N-terminus consists of a PWWP domain, which binds nucleosomes trimethylated at Lys36 of histone H3 (H3K36me3), an epigenetic mark associated with transcriptionally active sites [102,103]. Stringent knockdown or knockout of LEDGF/p75 significantly diminishes HIV-1 titers by specifically inhibiting integration, and also changes integration site-selection [104,105,106]. Additionally, replacing the PWWP domain of LEDGF/p75 with a heterologous chromatin binding domain redirects HIV-1 integration to chromatin regions bound by the alternative domain [107], further supporting the conclusion that LEDGF/p75 is responsible for guiding and tethering the HIV-1 PIC to its integration site.

Integrase strand-transfer inhibitors (INSTIs) prevent the integration reaction by targeting the strand transfer step [3]. These drugs bind to the active site of the IN CCD, displacing the reactive 3’ end of the viral DNA and preventing its insertion into the host DNA [73]. Mutations in the IN active site confer resistance to INSTIs by directly or indirectly inhibiting drug binding, albeit at a viral fitness cost (reviewed in [15,108,109]). As such, other compensatory mutations that increase the catalytic activity of IN are additionally found in patients undergoing INSTI therapy [108,109]. Emergence of resistance and cross-resistance is commonly observed for the two first-generation INSTIs, raltegravir and elvitegravir [110]. In spite of the improved potency and higher barriers for resistance, second-generation inhibitors also select for viral resistance [108,111], highlighting the need for antiretroviral compounds that inhibit IN by a different mode of action. 

## 4. HIV-1 Gag as the Master Regulator of Virion Assembly

Virion assembly, release, and maturation is a multistep process involving coordinated protein–protein, RNA–RNA, and protein–RNA interactions (reviewed in [112]). Like all retroviruses, HIV-1 selectively packages two copies of the full-length vRNA genome (reviewed in [113]), which are non-covalently dimerized at their 5’ untranslated region (5’ UTR). The HIV-1 5’ UTR is highly structured and forms six stem-loops with various roles in transcriptional regulation, reverse transcription, dimerization, RNA splicing, and packaging (reviewed in [114]). The regions responsible for RNA dimerization and packaging overlap and contain four stem loop structures, SL1, SL2, SL3, and SL4, which are often collectively referred to as the packaging sequence, or psi (Ψ). Dimerization of the RNA molecules is required for packaging and infectivity, and is initiated by a region termed the dimer initiation site in SL1. This site contains an apical bulge of nine bases, six of which form a palindrome, allowing the formation of classic Watson–Crick base pairs with the complementary sequence on the other RNA molecule, resulting in a “kissing-loop” structure [47,114,115,116]. The dimer initiation site is able to mediate dimerization of RNA molecules both in vitro [117,118,119,120] and in vivo [121,122,123], and is a major determinant in partner selection and copackaging [121,122].

Early biochemical studies have found that Gag first interacts with vRNA in the cytoplasm as a monomer or low-order multimer, and brings the genome to the plasma membrane [124]. Further Gag molecules are recruited to this nucleation site, and Gag forms high-order multimers through interactions mediated by CA–CA interactions with the neighboring Gag molecules. Many of these findings were later corroborated by total internal reflection fluorescence (TIRF) microscopy studies [125,126,127]. In these experiments, vRNA was observed reaching the plasma membrane first, followed by recruitment of further Gag molecules soon after. In the absence of Gag, vRNA moved rapidly towards the plasma membrane, suggesting that Gag is responsible for docking vRNA at the plasma membrane. Over time, the amount of Gag at the nucleation site increased, consistent with many Gag molecules polymerizing around the initial Gag–RNA complexes [125,126,127]. 

The main contact point with vRNA within Gag is its NC domain, which is later cleaved to form mature NC protein during virion maturation. The 5’ UTR SL2 and SL3 RNA structures appear to be recognized by NC, which adopts distinct conformations to bind either stem loop [128,129]. In addition to recognizing structured elements on the HIV-1 RNA, there is also evidence that Gag recognizes dimerized RNA [130]. While a minimal sequence both necessary and sufficient for the packaging of the HIV-1 genome has not been defined, an RNA sequence containing SL1, SL2, and SL3 can both dimerize and bind NC in vitro [131], and mutations within the 5’ packaging sequence prevent RNA being packaged into viral particles [132]. Likewise, deletion of NC prevents RNA from being packaged and generates particles devoid of the HIV-1 genome [133]. NC binding to RNA is mediated by two CCHC-type zinc knuckle motifs [134,135,136], and swapping the NC domain of HIV-1 Gag with that of the murine leukemia virus (MLV) Gag allows the chimeric HIV-1 Gag protein to package the MLV genome [137,138], highlighting the importance of NC in selective genome packaging. Interestingly, replacing the HIV-1 Gag NC domain with the mouse mammary tumor virus (MMTV) NC domain does not change Gag’s preference for packaging HIV-1 RNA [139], suggesting that NC alone does not account for the specificity of HIV-1 genome packaging. Gag-RNA binding is dynamic, and changes as virions assemble, bud, and mature. In the cytosol the Gag NC domain preferentially binds structured elements of the HIV-1 genome (i.e., Ψ and RRE) and displays a preference to bind G- and U-rich elements on cellular mRNAs, while the matrix domain (MA) selectively binds cellular tRNAs [140]. In contrast, during virion assembly at the plasma membrane, NC preferentially binds A-rich sequences on the viral genome as well as on cellular mRNAs, while MA dissociates from tRNAs and binds the plasma membrane, facilitating budding of the virion [140]. 

After assembling at the plasma membrane, spherical immature virions bud off from the infected cell (Figure 2). In immature particles approximately 2000–4000 Gag molecules [141] are radially arranged along the viral envelope, with the matrix domain (MA) anchored to the membrane at one end and nucleocapsid (NC), still bound to viral RNA, projecting towards the interior (Figure 2). Immediately after or during budding, the virion undergoes a maturation process in which the viral protease enzyme (PR) cleaves Gag and Gag-Pol at multiple sites in a defined sequence to produce independent viral structural and replicative proteins (Figure 2). Gag is cleaved to produce MA, capsid (CA), NC, p6, and two spacer peptides (SP1 and SP2) while Pol is cleaved to yield the viral enzymes, protease (PR), reverse transcriptase (RT), and IN (reviewed in [47,48,142]). The processed proteins then rearrange to form the structure of the mature virion. MA remains associated with the inner side of the viral membrane and forms a discontinuous shell immediately under the membrane (Figure 2). Approximately 1000–1500 monomers of CA assemble to form the capsid lattice [141]. In HIV-1 the capsid takes on a characteristic conical shape, and is composed of approximately 250 hexameric and 12 pentameric rings of CA that are stabilized by interactions within and between subunits [143,144,145,146,147,148]. Enclosed inside the viral capsid are the two single-stranded HIV-1 RNA molecules bound by NC, and associated with IN and RT, together forming the vRNP (reviewed in [47,48,142]).

Thus, virion morphogenesis is a highly complex process that requires coordinated interaction between the Gag polyprotein and viral RNA, as well as regulated cleavage of Gag into separate mature proteins. While the process has long been thought to be driven solely by Gag, there is emerging evidence that IN plays an unexpected role in proper placement of the viral RNA genome inside the capsid during maturation.

## 5. Role of Integrase in Particle Morphogenesis

While integration is the canonical function of IN, early mutagenesis studies indicated that IN may also play a role in other aspects of virus replication. In particular, a group of IN substitutions referred to as class II IN mutations lead to pleiotropic effects in HIV-1 replication, including defects in particle assembly [57,149,150,151,152,153,154,155,156,157,158,159,160,161], morphogenesis [18,57,151,157,158,159,162,163], and reverse transcription in target cells [18,56,57,153,155,156,157,159,161,162,163,164,165,166,167,168,169,170,171,172,173,174,175,176,177,178,179], in some cases without impacting IN catalytic function in vitro [55,151,152,155,156,165,166,169,171,180,181]. When visualized using electron microscopy, class II IN mutant viruses generate particles with vRNP complexes mislocalized outside the capsid lattice [18,57,151,157,158,159,162,163]. A similar phenotype was noted in IN-deleted viruses [158], again suggesting that IN is necessary for proper virion morphogenesis. Such aberrant viral particles are generally referred to as “eccentric particles,” due to the mislocalization of the vRNPs outside the capsid lattice, and are morphologically distinct from immature virions. 

Surprisingly, it was recently discovered that treatment of virus-producing cells with a class of 2-(quinolin-3-yl) acetic acid derivatives known as allosteric IN inhibitors (ALLINIs) (also called noncatalytic IN inhibitors (NCINIs), lens epithelium-derived growth factor (LEDGF)/p75-IN inhibitors (LEDGINs), IN-LEDGF/p75 allosteric inhibitors (INLAIs), or multimeric IN inhibitors (MINIs)) results in generation of particles with eccentric morphologies [162,163,182,183]. ALLINIs were originally designed to prevent integration by interfering with IN binding to the cellular cofactor, lens epithelium-derived growth factor (LEDGF/p75), important for targeting the viral pre-integration complex to the host chromosome [184]. The compounds compete with LEDGF binding to IN by engaging the V-shaped binding pocket created by the catalytic core domain of two IN dimers in the intasome complex [163,184,185,186,187,188,189]. In addition to preventing IN–LEDGF interactions, ALLINIs also prevent integration in a LEDGF-independent manner by inducing aberrant IN multimerization, locking IN in catalytically inactive multimers that are unable to assemble on viral DNA and carry out the integration reaction (reviewed in [185,189]). However, subsequent studies found that many ALLINIs are more potent when added to producer cells, and that they inhibit viral replication at the later stages of the viral life cycle [163,182,183,186,187,188]. Specifically, treatment with ALLINIs interferes with virion morphogenesis and leads to the generation of eccentric viral particles with vRNPs mislocalized outside the capsid lattice, strikingly similar to those generated by class II IN mutations [163,182,183,187,190]. Similar to the mechanism by which they can prevent integration, ALLINIs are proposed to interfere with virion morphogenesis by inducing aberrant IN multimerization, and mutations that confer resistance to ALLINIs also prevent ALLINI-induced IN multimerization [182,191]. Many class II IN mutations also alter IN multimerization [76,180,192,193], suggesting that proper multimerization is important for IN’s function during virion morphogenesis. However, a defined mechanism by which IN ensures viral RNA is correctly packaged inside the capsid lattice remained elusive for many years.

A seminal study in 2016 revealed that IN binds viral genomic RNA in mature virions, and that IN–RNA binding is necessary for viral replication [18]. Crosslinking immunoprecipitation sequencing (CLIP-seq), an approach that captures protein–RNA interactions in relevant physiological settings [140], was instrumental in this discovery and demonstrated that IN binds the HIV-1 genome at discrete sites with a distinct binding pattern from that of NC. IN not only binds RNA, but also modulates RNA structure in vitro by bridging multiple RNA molecules together [18]. Several basic residues in the IN CTD, K264, K266, and K273, directly interact with RNA, and substitutions at these positions abolish IN–RNA binding in virions. Importantly, virus production in the presence of ALLINIs, BI-D and BI-B2, also prevented IN–RNA binding, likely through aberrant IN multimerization as detailed below [18]. Finally, inhibiting IN interactions with RNA, either by introducing mutations at the CTD binding site or by ALLINI treatment, leads to the generation of eccentric, non-infectious viral particles (Figure 3) with vRNPs mislocalized outside of the core [18].

This discovery uncovered an important clue as to how IN contributes to virion morphogenesis by highlighting its interaction with RNA, but also raised questions regarding the relationship between IN–RNA binding, IN multimerization, and proper virion morphogenesis. Specifically, the findings described above strongly indicated that IN–RNA binding was a driving force behind proper virion maturation, but the observation that numerous class II IN mutations lead to eccentric virion morphology, yet are distally located from the RNA-binding site in the IN CTD, argued that the loss of IN–RNA may be correlative and another characteristic of IN, such as its proper multimerization, was the true determining factor.

Recent characterization of the effects of class II IN mutations on IN–RNA binding, IN multimerization, and virion morphology has revealed that all of the class II IN substitutions examined compromise IN–RNA binding and lead to the generation of eccentric viral particles [190]. IN–RNA binding was prevented by one of three distinct mechanisms: reducing IN levels in virions and precluding formation of IN–RNA complexes, directly preventing IN–RNA binding without substantially affecting IN levels or IN multimerization, but most commonly through adversely affecting functional IN multimerization and indirectly impairing IN–RNA binding. In vitro, IN binds RNA as tetramers, and class II IN mutant proteins that form predominantly dimers have a reduced affinity for RNA. The mutations cause an even greater defect in the ability of IN to bridge multiple RNA molecules, suggesting that although IN dimers may be able to weakly bind RNA, IN must form tetramers to bind RNA with high affinity and recruit additional viral RNA molecules, as would possibly occur when viral RNA is condensed and placed within the core during maturation. Taken together, these recent findings argue that proper IN multimerization is likely a prerequisite for IN–RNA binding in virions and is important to IN’s function in virion morphogenesis. The identification of multiple mechanisms responsible for the loss of IN–RNA binding helps answer how multiple IN substitutions can cause the same phenotype and strengthens the conclusion that IN–RNA interactions account for the key role of IN in virion morphogenesis [190]. 

Whether IN plays a similar role during virion maturation in other retroviruses has yet to be studied. Although the process of maturation is relatively conserved across different retroviruses, there is considerable diversity in particle and core morphologies [147,194,195]. Interestingly, mutations in the C-terminus of the murine leukemia virus (MLV) IN cause defects in reverse transcription reminiscent of those caused by class II IN mutations in HIV-1 [196,197]. More studies are needed to determine if this defect is the result of aberrant virion morphology, or if IN–RNA interaction is required for proper virion morphology.

## 6. The Fate of Eccentric Viral Particles

It is intriguing that eccentric viral particles are unable to complete reverse transcription in target cells despite packaging comparable levels of RT enzyme and vRNA as wild-type particles [162,198]. In addition, neither the binding pattern of NC [199] on the viral genome nor genome condensation by NC [162] seem to be affected in eccentric virions. Multiple studies have reported that class II IN mutant viruses or viruses treated with ALLINIs are defective for reverse transcription in target cells [18,56,57,153,155,156,157,159,161,162,163,164,165,166,167,168,169,170,171,172,173,174,175,176,177,178,179,182,183,187,188,200,201], but offer varying explanations for this observation. One possibility is that direct interaction between IN and RT is required for reverse transcription, a hypothesis that is bolstered by the finding that IN interacts with RT in vitro, and IN mutations that abolish this interaction also prevent reverse transcription in cells [175,201,202]. It is also possible that while vRNA is mislocalized outside the viral capsid in eccentric particles, RT is not, and physical separation of RT from vRNA underlies the defect in reverse transcription [199]. Finally, there is growing evidence that the defect in viral infectivity is caused by the premature loss of vRNA and IN in target cells. Several studies have found that when class II IN mutant viruses or ALLINI-treated viruses enter target cells, vRNA and IN itself is lost from the cells using both biochemical [199] and microscopy-based assays [203]. Interestingly, mutations in CA that destabilize the capsid lattice in vitro also block reverse transcription in target cells [28,29,30,31], and treating producer cells with a CA-targeting compound leads to the generation of eccentric viral particles defective for reverse transcription [204], much like viral particles produced in the presence of ALLINIs. Taken together, these findings argue that proper encapsidation within the viral core is necessary to protect vRNA and viral replicative proteins from the host cell environment, and when unprotected by the viral capsid, vRNA and IN are either passively or actively degraded.

A mechanism responsible for the loss of vRNA and IN in target cells has yet to be defined. It is possible that the high AU-content of HIV-1 vRNA makes it inherently unstable [205,206,207], in a manner similar to that of cellular mRNAs that encode cytokines and growth factors [208]. It is also possible that when ectopically expressed alone in cells, IN undergoes proteasomal degradation [209,210,211,212,213], and knockdown of a cellular component of the ubiquitin-conjugation system, E3 RING ligase TRIM33, enhances HIV-1 infection and replication in cells [210]. However, another study found that during infection with eccentric viral particles, the loss of vRNA and IN was proteasome-independent [199]. Future studies are warranted to determine the precise mechanism of vRNP degradation upon its premature exposure.

In addition to preventing premature degradation of vRNA and IN, the CA lattice may also play an important role in shielding viral core components from host immune recognition. Mutating the HIV-1 CA to prevent its interaction with cellular factors leads to activation of innate immune responses and cytokine production in infected dendritic cells [214] and activation of a type 1 interferon response in infected monocyte-derived macrophages [215], suggesting that the CA lattice is important for recruiting cellular proteins to cloak viral components from detection. While it is important to note that infected dendritic cells appeared to detect reverse-transcribed viral DNA through the cytosolic DNA sensor cGAS [214], innate cytokine expression was also elicited in peripheral blood mononuclear cells transfected with purified HIV-1 RNA [216], demonstrating that HIV-1 RNA does have the potential to trigger immune activation. 

## 7. Therapeutic Outlook and Conclusions

Currently, all clinically approved IN inhibitors share a common mode of action and target the strand-transfer reaction during integration, making viral cross-resistance a problem. Therefore, there has long been interest in targeting alternative functions of IN, and its vital role during virion morphogenesis has become an attractive drug target. While not initially designed to interfere with viral particle maturation, many ALLINIs lead to the generation of eccentric viral particles in addition to inhibiting integration [18]. This dual-mode of action allows these compounds to retain antiviral activity even when viral resistance develops to one mode of action [217], and gives ALLINIs a distinct and non-overlapping resistance profile with INSTIs [218,219]. While resistance mutations do arise to ALLINIs, they are accompanied by a viral fitness cost. Importantly, mutations that confer resistance to ALLINIs can disrupt virion morphogenesis themselves, and compensatory mutations are required to overcome this defect [220], indicating a high barrier to resistance. Several compounds are currently in clinical trials after demonstrating favorable bioavailability, tolerability, and pharmacokinetics, and with more in development, ALLINIs are a promising new class of antiretrovirals.

CA has also emerged as a potential target for compounds designed to interfere with virion morphogenesis. Compounds that disrupt the stability of the capsid lattice and cause morphological defects in virions also block viral replication at or prior to reverse transcription [204,221], and they likely lead to the premature loss of vRNA and IN similar to how ALLINIs work. Two recent studies reported the effectiveness of a CA-targeting small molecule compounds in vivo, both in a humanized mouse model [222] and in humans [223]. Both compounds demonstrated potent and sustained antiviral activity in vitro, with no measurable cross-resistance with other antiretroviral drugs [223]. While it is important to note that these compounds led to morphological aberrations in virions by accelerating CA assembly rather than destabilizing it and appear to exert their antiviral effects primarily by preventing the nuclear import of viral DNA, they still highlight the importance of proper capsid assembly to viral replication and the value of interfering with virion morphogenesis as a therapeutic strategy.

In conclusion, virion morphogenesis is a critical step in the HIV-1 life cycle, and the discovery that IN plays a key role in this process opens up new doors for therapeutic interventions. IN is crucial for ensuring that the viral RNA genome is packaged inside of the capsid lattice in virions, and interfering with this function of IN leads to morphological defects that prevent further viral replication. Small molecule compounds that exert this effect can complement existing antiretroviral compounds already in the clinic, and when used in combination with INSTIs could further raise the barrier to drug resistance. Destabilizing the viral capsid by targeting CA has similar effects on viral replication, and is also a viable therapeutic strategy. A better understanding of the events surrounding virion morphogenesis will be important to help guide the design of future therapeutics.

## Figures and Tables

**Figure 1 viruses-12-01005-f001:**
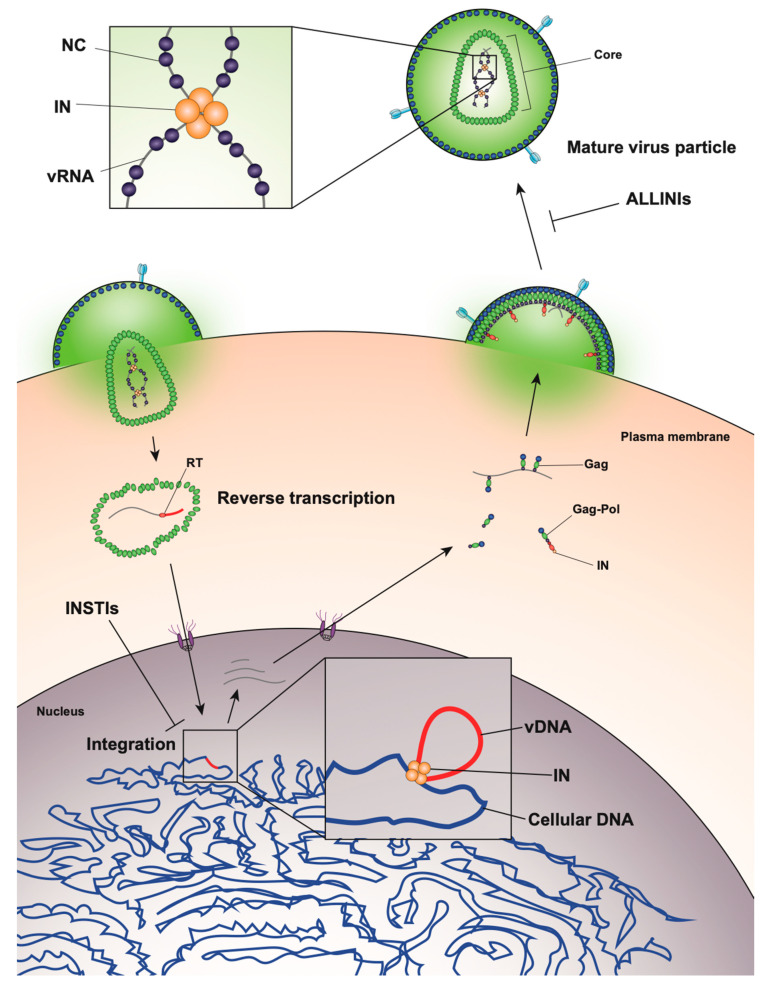
Overview of the human immunodeficiency virus type 1 (HIV-1) life cycle highlighting the key role of HIV-1 integrase enzyme (IN) at early and late stages of replication.

**Figure 2 viruses-12-01005-f002:**
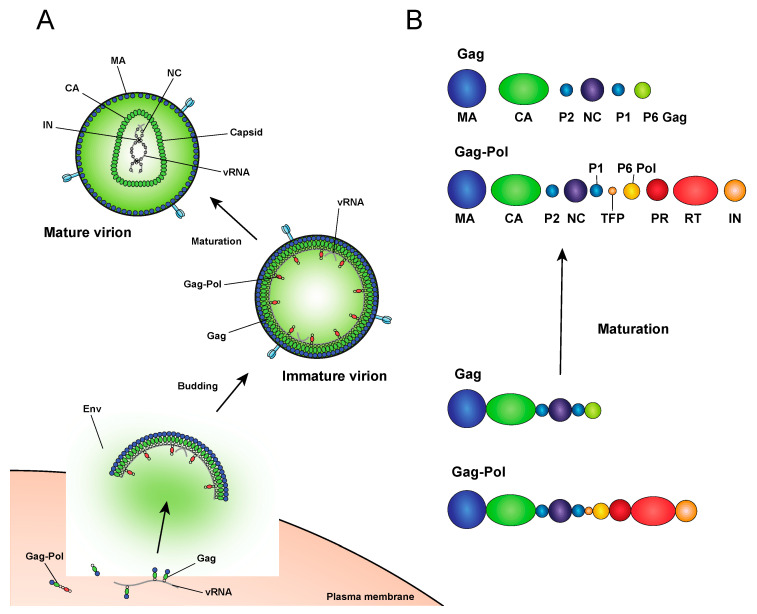
Virion maturation and morphogenesis. (**A**) The Gag and Gag-Pol polyproteins assemble with vRNA at the plasma membrane, bud from the surface of the cell as immature virions, and then undergo a maturation process. (**B**) During maturation PR cleaves Gag and Gag-Pol into independent structural and replicative proteins.

**Figure 3 viruses-12-01005-f003:**
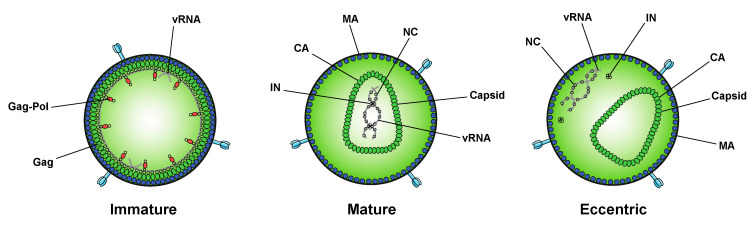
Inhibition of IN–RNA interactions result in the formation of eccentric virions. Immature viral particles consist of many molecules of Gag and Gag–Pol concentrically arranged along the inner leaflet of the viral membrane and bound to viral RNA (vRNA) at the NC domain. In mature viral particles, the vRNA is bound by NC and condensed with RT and IN to form the vRNP, which is enclosed in the conical capsid lattice made up of CA monomers. In eccentric viral particles, as observed upon inhibition of IN binding to the viral genome, the vRNP is mislocalized outside of the capsid lattice.

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
