# Peer review of "Going beyond Integration: The Emerging Role of HIV-1 Integrase in Virion Morphogenesis"

_viruses, 2020, doi:10.3390/v12091005_

Round 1

Reviewer 1 Report

In this review, Jennifer L. et al., provided an overview of multiple roles of integrase (IN) in the HIV-1 life cycle, with a focus on its function during virion maturation and morphogenesis. In general, the manuscript is well organized to catch-up the recent accumulating information of the IN function(s) during virion morphogenesis. The content is suitable to be published in the section of “Function and Structure of Viral Ribonucleoproteins Complexes”. I would suggest some points, however, to be considered or edited before publications.

Specific comments.

  • Some detailed description not directly related to IN function(s) in the “Overview of HIV-1 life cycle” could be simplified. These include lines 66-68, lines 86-89 and lines 95-98.
  • Fig. 1: I would suggest to depict the target sites by INSTIs and ALLINIs to highlight the IN key functions. Also, please indicate which portion represents the viral “core” in Fig.1.
  • Lines 35 and 82: Proviral DNA was the term for integrated form of viral DNA. I would suggest to edit “proviral DNA” to “viral DNA” or “viral DNA copy”.
  • Line 354-355: I would say that Ref188 did not tested all of the class II IN mutations published before, so please modify the term of “without exception”, or replaced with “all of class II IN substitutions examined”.
  • Line 364: Please clarify “additional RNA molecules”, cellular or viral RNAs?
  • Line 389: Alongside with refs 173 &199, please cite the recent paper by Takahata et. al., which showed that IN functions during reverse transcription through RT-IN precursor form ( Virol. 91, e02003-16, 2017, doi:10.1128/JVI.02003-16)

Author Response

We thank the reviewer for the useful suggestions and comments. We have incorporated all the proposed changes in the manuscript and simplified the text as recommended. Below are point by point responses:

  • Fig. 1: I would suggest to depict the target sites by INSTIs and ALLINIs to highlight the IN key functions. Also, please indicate which portion represents the viral “core” in Fig.1.
    • Response: We have incorporated this change in the Figure.
  • Lines 35 and 82: Proviral DNA was the term for integrated form of viral DNA. I would suggest to edit “proviral DNA” to “viral DNA” or “viral DNA copy”.
    • We have incorporated these changes at line 35 and line 95 of the new manuscript.
  • Line 354-355: I would say that Ref188 did not tested all of the class II IN mutations published before, so please modify the term of “without exception”, or replaced with “all of class II IN substitutions examined”.
    • We have incorporated this change in lines 403-404 of the current manuscript.
  • Line 364: Please clarify “additional RNA molecules”, cellular or viral RNAs?
    • We clarified this and changed it to "viral RNA" in line 413 of the current manuscript.
  • Line 389: Alongside with refs 173 &199, please cite the recent paper by Takahata et. al., which showed that IN functions during reverse transcription through RT-IN precursor form ( Virol. 91, e02003-16, 2017, doi:10.1128/JVI.02003-16)
    • We have incorporated this reference in line 439 of the current manuscript.

Reviewer 2 Report

This is a very nice review and it is a joy to read. There are some minor suggestions:
1. Can the authors draw the nuclear pore complex on Figure 1 and elaborate how HIV RNA/protein complex goes in and out of the nuclear pore?

2. Since the title mentioned Integrase in Virion Morphogenesis, can the authors show in illustrations on morphogenesis in figure 1, such as eccentric viral particles and/or other relevant morphological changes?

Author Response

We would like to thank the reviewer for his/her positive feedback and useful comments on the manuscript.

1. Can the authors draw the nuclear pore complex on Figure 1 and elaborate how HIV RNA/protein complex goes in and out of the nuclear pore?

Response: We highlighted the nuclear pore in Fig. 1 and referenced two recent review articles which in detail explains the molecular factors that mediate entry of these complexes in lines 93-95 of the current manuscript.

2. Since the title mentioned Integrase in Virion Morphogenesis, can the authors show in illustrations on morphogenesis in figure 1, such as eccentric viral particles and/or other relevant morphological changes?

Response: We decided to add 2 additional Figures (Figure 2 and 3) to highlight these points.